# A Conceptual Framework for Analyzing Social Representation in Unstructured Data

## Abstract

Unstructured data used in foundation model development is a challenge for systematic analyses to make data use and documentation decisions. From a Responsible AI perspective, these decisions often rely upon understanding how people are represented in data. We propose a framework to guide analysis of human representation in unstructured data and identify downstream risks. We apply the framework in two toy examples using the Common Crawl web text corpus (C4) (Raffel et al., 2020), and LAION-400M (Schuhmann et al., 2021). We also propose hypothetical action steps in service of dataset use, development, and documentation.

## 1 Introduction

Data is recognized as a core underlying factor contributing to machine learning model behaviours that can be unfair or harmful to humans (Paullada et al., 2021). Insights from systematic analysis of datasets can identify potential harms and inform interventions to mitigate risk. Principled analysis of the data underpinning pre-trained foundation models is particularly salient given the increasing reach of such models and their use by researchers and developers who lack the resources to develop computationally-intensive models (Bommasani et al., 2021; Han et al., 2021).

At the same time, the large, unstructured nature of these datasets poses significant challenges for conducting analyses required to make development, documentation, and use decisions. The open-ended potential for downstream use, means that risks are wide-ranging and sometimes lack clear methods of evaluation (Weidinger et al., 2021). Prior systematic fairness audits have often focused on data labels and utilized aggregated and disaggregated analyses to identify class imbalances (e.g., (Saleiro et al., 2018; Kearns et al., 2018; Kleinberg et al., 2016; Friedler et al., 2019)). Despite increased scrutiny of large unstructured datasets (Birhane et al., 2021; Dodge et al., 2021), methods of analysis remain less robust and less systematic relative to labeled datasets, in part because labels provide a crucial pointer to dataset features to evaluate for fairness and bias concerns.

We close this gap by contributing a conceptual framework to standardize workflows for analyzing unstructured data. The framework (shown in Appendix A) focuses on social representation of people in data, including the data features that indicate social identity and influence the representation of different social groups.While many evaluations of unstructured data exist across ML, there is little guidance or structure for applying them in practice to fairness workflows. As a result, practitioners apply analyses ad hoc, continue to use what they have used before, or miss relevant analyses (Madaio et al., 2022; Heger et al., 2022). Our primary contribution is a conceptual structure– that is, a sociotechnical organization of analyses which are grouped according to *who* is in the data, *what* is in the data, and associations between the two. Thus, the structure and core analytical questions are modality-agnostic and extensible to new modality combinations. The framework does not strictly prescribe analysis implementations– rather, it guides responsible AI (RAI) workflow planning for data evaluation, documentation, and risk mitigation.

## 2 Background

### 2.1 Dataset Transparency and Documentation

A growing body of scholarship in RAI focuses on increasing transparency of AI systems and datasets for a variety of stakeholders. These range from developers who build on pre-trained models to system

end-users who may be subject to algorithmic decision making (Lima et al., 2022; Wagner et al., 2020). At the dataset level, transparency highlights critical information about the contents of a dataset as well as the processes that underpin how a dataset was created. To this end, a range of work brings structured approaches to documenting both dataset content and development processes (Bender & Friedman, 2018; Gebru et al., 2021; Dodge et al., 2021; Díaz et al., 2022; Hutchinson et al., 2021; Rostamzadeh et al., 2022; Srinivasan et al., 2021; Pushkarna et al., 2022). However as massive, unstructured datasets increasingly become the norm in ML development, structured frameworks are needed to help summarize key characteristics of the data they capture. Dodge et al. (2021) offer an expansive audit of C4 (Raffel et al., 2020), which inspires a more structured approach for disentangling the contents of web-crawled data that feature heavily in ML datasets. Our work aims to standardize approaches to support existing transparency and documentation efforts by enabling the identification and communication of potential social risks associated with data.

## 2.2 DATASET AUDITS

Datasets underpinning training and testing have been at the center of various tensions connected to privacy, consent, unfair system performance, representational harms, and harmful applications (Paullada et al., 2021). Against this backdrop, prominent ML datasets have been subject to close scrutiny, with empirical examinations and audits uncovering a range of problematic content that itself is harmful (e.g., copyright violations; representational harms such as misgendering) or that can lead to downstream harms. For example, both image and text datasets have been shown to contain co-occurrence statistics that mirror harmful social stereotypes (Garg et al., 2018; Hendricks et al., 2018); image datasets have been found to include problematic sexual imagery, including depictions of sexual violence and non-consensual sexual content, and racial and ethnic slurs within image labels and captions (Birhane & Prabhu, 2021; Birhane et al., 2021); Dataset audits can close documentation gaps (Dodge et al., 2021), be used to make data filtering or re-balancing decisions (Russakovsky et al., 2014), and, in some extreme cases, lead to the deprecation of datasets, such as MegaFace (Kemelmacher-Shlizerman et al., 2016) and Tiny Images (Torralba et al., 2008). Organizing prior individual audits, we present a principled framework that supports dataset auditing to both shape dataset development decisions, as well as flag downstream model evaluations to prioritize.

## 2.3 STANDARDIZING RESPONSIBLE AI WORKFLOWS

Evaluating data in a structured way and communicating results to stakeholders remains an important challenge for RAI. Data work often takes a backseat to work focused on developing state of the art models and algorithms (Sambasivan et al., 2021b). In addition, current approaches to data documentation are "largely ad hoc and myopic in nature" (Heger et al., 2022) and practitioners face difficulty in understanding why documentation is needed, how best to document, and, ultimately, what to document (Chang & Custis, 2022). A range of development toolkits and checklists have been proposed to address these challenges, including documentation frameworks such as Data and Model Cards (Gebru et al., 2021; Mitchell et al., 2019; Pushkarna et al., 2022), internal auditing frameworks (Raji et al., 2020b), and impact assessment frameworks (Schiff et al., 2020). RAI audits in particular require support to determine what to measure and how to measure it to avoid risks that can compound through development (Sambasivan et al., 2021b). Mitchell et al. (2022) give a high-level framework for measuring large, unstructured datasets and we extend this by configuring our framework around downstream risks and demonstrate how the results of an audit can be used to distill dataset decisions.

## 3 FRAMEWORK

In this section, we introduce a framework for systematic evaluation, anchoring on risk and harm associated with social representation in data. The full framework, including a list of data analyses and dependencies can be found in the Appendix. The framework supports analyses for a variety of goals ranging from dataset development to third-party audits. Our framework also identifies a set of components that guide the operationalization of each analysis and the interpretation of their results.

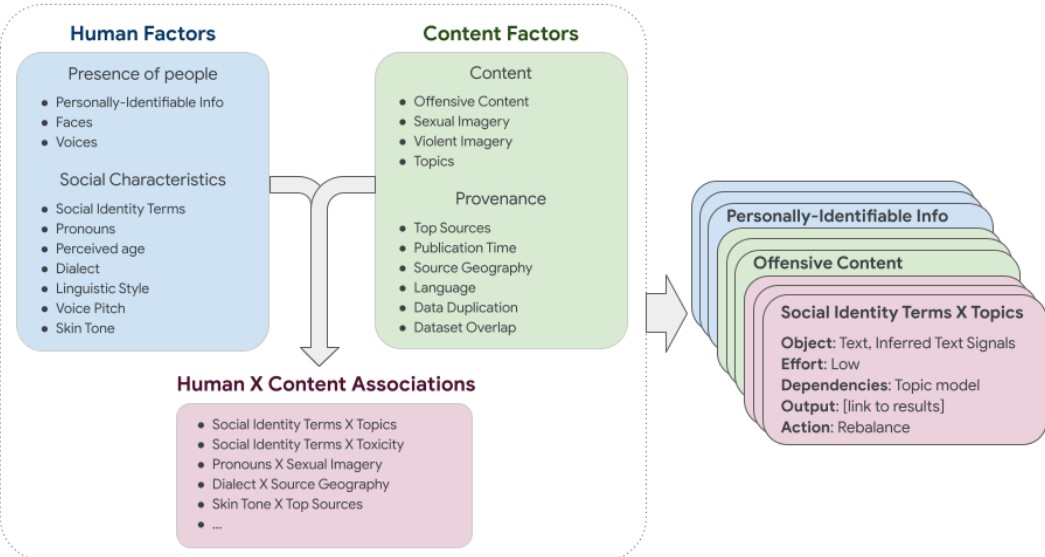

Figure 1: Framework conceptual structure. Guiding and output components are shown on the right.

## 3.1 FRAMEWORK ANALYSES

Figure 1 demonstrates the framework's conceptual structure. We organize the framework around high-level questions about human-centered considerations in data: namely, *Who* is in the data, *What* is in the data, and *How* are the two associated? This structure also allows the analyses to focus on data questions at different levels of complexity with respect to corresponding downstream harms, as well as to prevent an over-focus on optimizing isolated analyses or metrics.

Our framework is designed to be general-purpose and extensible, thus it is not exhaustive of every single possible analysis within each section. While we use text, image, and image-text datasets as references for developing and describing the framework, it can be adapted to other modalities with appropriate changes. For instance, the dialect analysis in text could be adjusted to include elements of (or complemented with) analysis of accent in speech data. Given that SOTA implementations of an analysis will change periodically, we focus instead on the goals of and workflows for generating and interpreting analysis results. Analyses can also be modified, added, or removed as the field's collective sociotechnical understanding about relevant social biases evolve over time, while preserving the overall framework structure. For example, salient social identity term lists may iteratively change as best practices respond to social shifts, or as global socio-cultural contexts are increasingly integrated into RAI considerations. Analyses can also be updated alongside our understanding of salient social risks, the human social characteristics they are connected to, and our technical means of analyzing them. However, the motivating questions remain stable.

### 3.1.1 WHO IS IN THE DATA?

In asking who is in the data, we consider several human factors of data that include measuring the presence of people in data along with social characteristics.

**Presence of People:** These analyses tally whether individuals or identifying information appear in data. This includes calculations of personally-identifiable information and face or person detection. Extending to new modalities, the analyses implicitly ask which data characteristics can indicate the presence of a person, such as faces or bodies in visual data or voice in audio data. Results guide more focused, follow up analyses that assess depictions of social groups.

**Social Characteristics:** These analyses center on data characteristics that are often associated with social identity and may be used as proxies for social identity. Some proxies appear directly in data, such as pronouns, while others, such as perceived age or gender expression in images, must be inferred, frequently using predictive methods (e.g., Lanitis et al. (2004)). These include analyses of

dialect, linguistic style, skin tone, and voice pitch. Social characteristic analyses provides insight into the over- and under-representation of specific social groups, which has been associated with disparities in performance (Wilson et al., 2019; Buolamwini & Gebru, 2018) and general problems for class prediction (Johnson & Khoshgoftaar, 2019). Because these characteristics are social in nature, their measurement must be adapted to local context and time. For example, social identity terms vary across social and cultural contexts, meaning static identity term list cannot be exhaustive.

### 3.1.2 WHAT IS IN THE DATA?

The second grouping of analyses focuses on content that may influence human representation.

**Content:** This group of analyses is focused on content characteristics that relate to harmful or undesirable outcomes that are independent of specific people or social groups. Analyses include calculating the distribution of topics in text, as well as sexual content in images. Topic distribution provides a birds-eye view of the composition of the data and can give an indication of sexually explicit or sensitive topics contained in a dataset. Topic distributions can give clues to subtle downstream biases. For example, models trained primarily on news data have been shown to exhibit biases against particular country names and professions (Huang et al., 2019).

**Provenance:** Data provenance can indicate the values, norms, and perspectives likely to be contained in data and ascertained through metadata, such as the geographic distribution of sources and their publication dates. For example, source URLs point to the range of content represented in web-scraped data, which offers insight into document content, such as linguistic and cultural content, as well as the prevalence of machine-generated text (Dodge et al., 2021). The geographic, cultural, and social representation in data can have implications for downstream models. For example, image classifiers trained on datasets sourced predominantly from western countries have lower rates of accuracy when applied to images from non-western countries (Shankar et al., 2017). Data recency can have particular impacts on models supporting low-resource languages, which can disproportionately rely on religious or historical texts due to data scarcity (e.g., Ahmadi & Masoud (2020)).

### 3.1.3 HUMAN × CONTENT ASSOCIATIONS

The final section focuses on associations between human and content factors, which reveal *how* people are depicted. Associations disaggregate analyses within and across modalities, such as social identity terms and topics in text or occurrences between objects detected in images and identity terms in associated text in multimodal datasets. Associations can reveal stereotype-aligned correlations, which can amplify the stereotypes and propagate exclusionary norms (Dev et al., 2021; Weidinger et al., 2021; Zhao et al., 2018; Hendricks et al., 2018). While highly specific combinations of analyses can be run (e.g., an evaluation of queer depictions in Spanish-language medical literature from a specific year), the structure of the framework facilitates analyses beginning with the most general question (i.e., are people depicted?) followed by more specific inquiries (e.g., with which other data do people most often occur?) to provide a tractable entry point for RAI analyses.

## 3.2 FRAMEWORK COMPONENTS

Next, we outline additional framework components that guide analysis results reporting and general analysis planning. The **Output** and **Action** fields are provided to capture the results of a given analysis and any mitigation actions decided in response. The Taking Action section discusses in more depth the process of making mitigation decisions. In addition to a research-backed motivation related to downstream risks, each analysis includes additional fields to support planning:

**Analysis Object** indicates whether an analysis is calculated on data directly (i.e., tokens in text data) or if it applies to an inference produced by an intermediate classifier (e.g., inferred document topic; predicted age of person in an image). This highlights which analyses are dependent on predictive models and therefore susceptible to biases that those models may themselves exhibit. The distinction between "Image" and "Inferred image signals" is particularly important since few analyses in the framework are applied to image data directly.

**Effort** indicates rough time and cost of an analysis based on current techniques and tooling, which, reflect bias toward use for English and Western data. In non-English and non-Western contexts, effort is often higher for implementation.

| Analysis Goals | |
|---|---|
| *Dataset Development* | Developing a dataset for training or evaluation through new data collection and/or adaptation of existing datasets |
| *Use Decisions* | Making decisions regarding appropriate use of a dataset, whether for training or evaluative purposes |
| *Model Understanding* | Investigating potential roots of or explanations for model behavior |
| *Auditing* | Auditing a dataset to fill documentation gaps, ensure legal or institutional compliance, or to foster greater public awareness |

Table 1: A non-exhaustive list of data analysis goals.

**Dependencies** indicates intermediate resources needed to conduct an analysis, classifiers which produce inferred signals. While the framework does not dictate a single, required implementation for any analysis, we point to example classifiers and term lists that may be used. Moreover, some dependencies, such as term lists, should ideally draw from qualitative insights to localize evaluations.

## 4 TAKING ACTION

The framework is not meant to be exhaustively implemented for every use case, since all possible association analyses would produce an intractably large number of results. Moreover, which mitigation actions to take depends on context; the downstream effects of data filtering or rebalancing are, in many cases, still an open question. Finally, while this framework can be used to discover new biases, it is intended to help practitioners overcome challenges in applying existing evaluations and mitigations motivated by institutional policies and well-documented data biases, such as gender bias. These challenges include needed guidance for identifying risks and harms AI systems can generate, and a systematic approach for applying known fairness evaluations across a broad range of products and systems (Madaio et al., 2022; Heger et al., 2022).

An exhaustive review of mitigation actions is beyond the scope of this work, however we describe key considerations that inform the actions a user should take. We also include a selection of guiding questions and considerations at the start of the full framework. Key questions narrow the scope of actions to be considered, making planning more tractable:

- What are the planned deliverables of the data effort (e.g., training or evaluation data)?
- What are the primary goals the analyses will support (e.g., making use decisions)?
- To what extent can development steps be revisited or modified (e.g., data collection, documentation)?
- To what extent is the dataset mutable (i.e., can data be added, filtered, or modified)?

**Dataset Purpose:** The framework can be applied to a range of dataset types including pre-training or fine-tuning datasets. It can also be used for understanding and evaluating model-generated data. Suitable framework actions depend on the purpose of the dataset. For example, when analyzing pre-training data, it may be unclear how changes to data distributions will impact model performance, potentially making other mitigations more desirable. In contrast, data used for benchmark development stands to be used as a repeated measure of model robustness and performance. Thus, actions that might require additional costs or resources may be more easily justified to meet evaluation goals.

**Analysis Goals:** A range of goals can motivate framework use– each of which brings attention to different actions. Table 1 lists common goals of dataset analyses. For example, developing a new dataset from web-scraped data raises potential decisions to collect additional data or adjust filtering criteria in the data collection process. In contrast, conducting an audit of a third-party dataset for compliance purposes brings focus to documentation and data use decisions.

**Development Phase:** Each development phase affords different actions to address data concerns. Table 2 shows common actions by development phase. For example, during data collection, toxic content biased across social identity groups might be addressed by modifying the dataset or by adjusting model evaluation planning. Alternatively, documentation can flag concerns for public consumption. Moreover, concerns may be addressed through explicit dataset release decisions.

The decision to pursue an action such as the ones listed above will depend on analysis goals, available resource play a key role in determining the mitigation actions that are available. Importantly, dataset

| Dev. Phase | Actions | Description |
|---|---|---|
| Data Collection/ Processing | Addition | Rebalancing distributions across an entire dataset or within specified categories with additional (potentially synthetic) data |
| | Removal | Filtering data to remove unwanted content |
| | Augmentation | Augmenting data, such as through data tagging (Anil et al., 2023) to allow a model to learn undesirable content while controlling its production downstream |
| | Flagging | Flagging analysis results for further downstream evaluation or documentation |
| | Non-Use | Not using the dataset, for example if applying analyses to different candidate datasets to decide which to use |
| Model Evaluation | Addt'l Benchmarking | Selection of additional evaluation benchmarks |
| | Benchmark Creation | Development of benchmarks to evaluate new concerns |
| Documentation | Warning | Documentation of general or use case-specific limitations |
| | Non-Use | Documentation of cases where the data should not be used |
| Packaging and Release | Licensing | Development of licensing and terms of use specifications |
| | Access | Development of limited access policies |

Table 2: A non-exhaustive list of actions that may be taken to address social risks identified in data.

actions such as filtering may exacerbate existing imbalances or introduce new ones, requiring iterative evaluation. Direct action also may not be possible, for example if there are cost constraints or if data sources and filtering techniques used to develop a third-party dataset are not clearly defined or known.

# 5   APPLYING THE FRAMEWORK

In order to showcase how the framework guides evaluations, we provide two toy examples using C4 (Raffel et al., 2020) and LAION-400M (Schuhmann et al., 2021)– two large, unstructured datasets available under a CC-BY 4.0 license. We apply analyses from the standpoint of a team seeking to repurpose data for their own use. Our **goal** is to develop derivative datasets from C4 and LAION-400M while assessing representational biases that have been broadly identified in text and image datasets. Our examples focus on known biases and they are not meant to uncover scientifically novel results, but rather provide a demonstration of how a practitioner can use the framework to meet analysis goals. For example, a range of gender biases in text datasets has already been discovered, however researchers and practitioners must conduct audits for generally known issues and comply with project specifications with each dataset they work with. Thus, when faced with the question, "how should I begin to evaluate gender bias in a dataset", this framework establishes a starting point.

We do not present every possible analysis; instead, we focus on a few key results. We do this for two reasons. First, not all analyses are relevant for understanding a specific social group or modality. Thus, in practice, only a selection of analyses will be conducted. Second, some analyses are not yet technically feasible and are themselves the subject of research (e.g., detecting hateful symbols and memes in images (Mathias et al., 2021)). Finally, because we primarily focus on how to use results across analyses to inform risk mitigation, we do not delve into technical details or performance metrics for the classifiers we use.

## 5.1   EVALUATING AGE REPRESENTATION IN C4

Prior research identifies age bias as an issue for ML and AI development (Díaz et al., 2018; Garcia de Alford et al., 2020) and calls for increased age representation in AI datasets (Park et al., 2021). With this in mind, we turn to assessing how older adults are depicted in C4 using Association analyses. We assess the results of age depictions via top-associated tokens and topics with age-related terms.

**Output:** We see in Figure 2 the tokens most associated with old age terms, which occur 110,000 times in the dataset. These include dementia, and degeneration, both of which can render negative sentiment. We see related associations in the topics disproportionately associated with old age terms. These include health topics, including medical conditions and assisted living, as well as skin and face care beauty products, which likely point to content covering anti-ageing products and discussion.

**Action:** In line with prior work, we find limited and skewed representation of older age. Work has been conducted on decoupling adjective associations from select identity terms (Dev et al.,

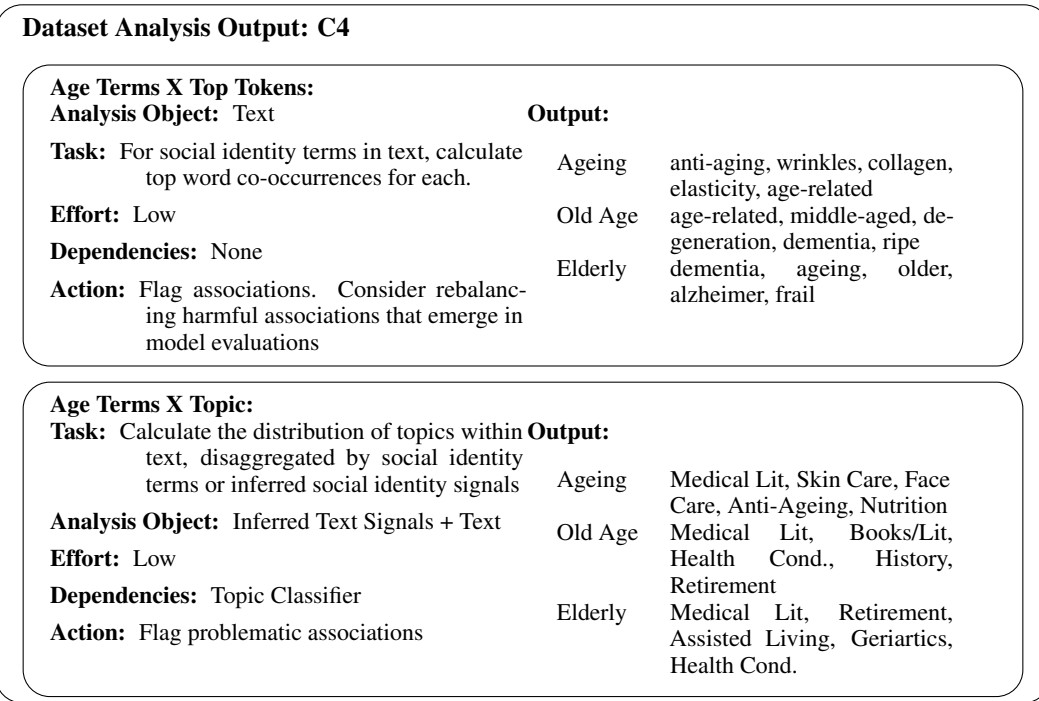

Figure 2: Sample dataset analyses output for C4.

2021), however broader sentential context surrounding age-related terms may still carry negative or stigmatized sentiment. Filtering or removing data stands to worsen older adult underrepresentation in ML datasets, however it is among the lowest cost options. For developing training data, other actions may be taken. If a data collection or generation pipeline is feasible, targeted data collection or synthetic data can be used to rebalance the data. Results can also be flagged to evaluate for similar biases in data generated by the downstream model. Contingent on these evaluations, documentation warnings or non-use in certain cases may also be necessary.

## 5.2 Evaluating Queer Representation in LAION-400M

LAION-400M features over 400 million image-text pairs extracted from Common Crawl. The dataset is unstructured and uncurated, though it does feature NSFW tags, which were used to identify a number of illicit images. Text-image datasets are shown to produce various social biases, such as gender and skin tone biases (Cho et al., 2022). Researchers have found undesirable associations with queer identity terms in text datasets (Dixon et al., 2018) and sexually explicit depictions of women in LAION-400M (Birhane & Prabhu, 2021; Birhane et al., 2021). While explicit content can be used for specific applications, unintentional inclusion risks unwanted generation of explicit content by downstream models. We assess a combination of these biases by evaluating queer representation and consider mitigations for adapting third-party training data. We run Association analyses in text using the same topic classifier from our prior analyses, and run multimodal Association analyses using a classifier similar to (Google Cloud Vision), which identifies sexual content in images. We use webref entities to obtain queer identity terms.

**Output:** Figure 3 shows the top topics associated with a variety of sexual identities. Prominent among these are those that seemingly refer to various sexual topics and activities. This includes for "heterosexuality". Interestingly, the most frequent topic for "heterosexuality" is "LGBT Porn" which suggests that the term is connected to a subgenre of pornographic videos. Though not a sexual orientation, the generally derogatory term "transsexual" is also strongly associated with sexual topics.

**Action:** Considering dataset usage for T2I training in particular, there is likely a very limited set of use cases in which the generation of sexual content would be appropriate. Such use cases would likely entail very specialized dataset curation and model development. Therefore, one could consider filtering sexual content in order to both limit its downstream production as well as to avoid the

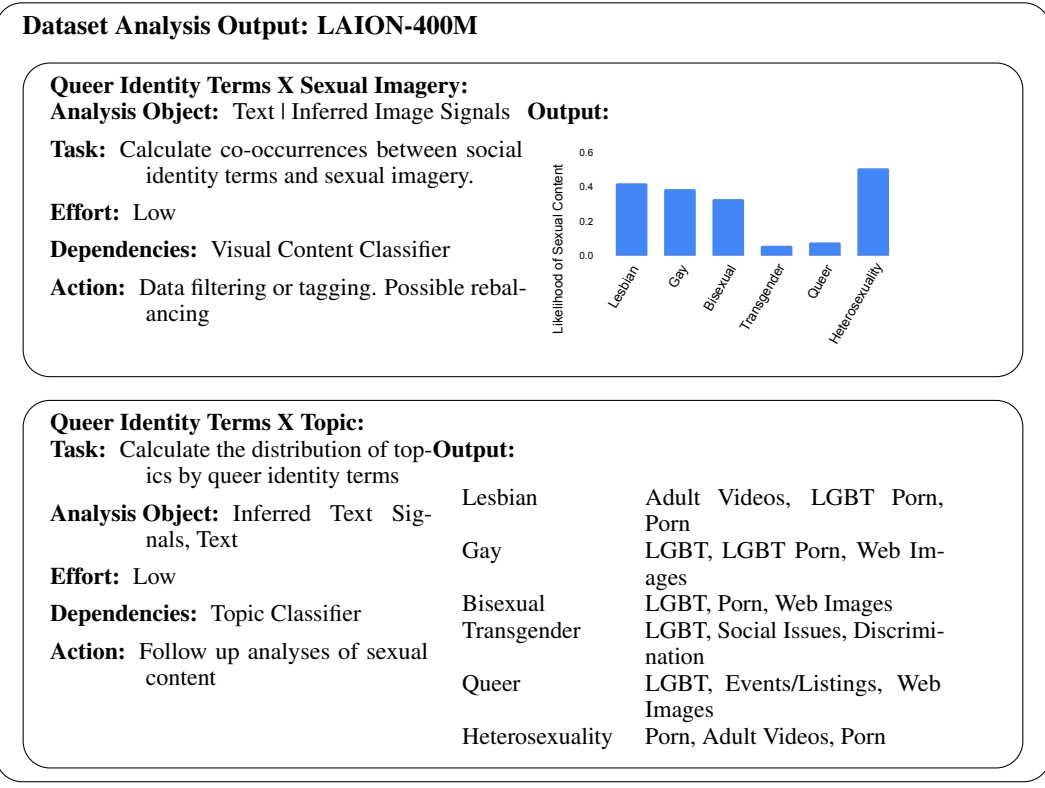

Figure 3: Sample dataset analyses output for LAION-400M Dataset.

inclusion of published sexual content, which has often been made public without sex worker consent (Cole, 2020). Because filtering may lead to removal of nearly half of the instances of some identity terms, rebalancing may also be needed. Alternatively, sexual content in text data could be augmented with tags to preserve a downstream model's ability to detect it while limiting its production.

## 6 EVALUATING REPRESENTATION IN DATA

In line with Mitchell et al. (2022)'s call to establish practices for measuring data, characterizing how people are represented in data is a necessary part of identifying risks. Yet, RAI lacks systematized guidance to do so across data modalities. In RAI work, there has been little guidance for using combinations of analyses and data features to measure latent representations of social identities.

Notably, our framework has no canonical list of social identities nor an exhaustive list of evaluations for a given social identity. This is because social identity is unstable in nature (Hanna et al., 2020), and the axes along which discrimination occurs are culturally specific (Sambasivan et al., 2021a). Characteristics associated with social identities change with context and the same features can be associated with disparate groups (e.g., hispanic surnames prevalent in both Latin America and the Philippines). Moreover, the semantic meaningfulness of data across modalities varies. Social identity terms can be easily identified in text; however image data, relies more heavily on labeling for automated evaluation. Other modalities, such as sensor data, may not have clear social signals.

Analyzing how people are depicted is also challenging because "good" social representation changes with context. As Chasalow & Levy (2021) posit, representativeness is both time and place specific. The social categories we attend to are shaped both by normative assumptions about what should be measured as well as the existence of a name for or conception of a social category. For example, Andrews et al. (2022)'s research suggests that a word list generated today to analyze disability representation in a dataset would likely feature different terminology than a list generated 30 years ago. Localizing analyses to evaluate specific communities and contexts should ideally be done through approaches that engage qualitative and participatory methods. Our framework is adaptable to analysis implementations that are localized to contextualized social identity cues.

### 6.1 THE ROLE OF DATASETS IN ASSESSING HARM

In developing this framework to support development decisions, we extend the work of others advocating for more attention to data work (Sambasivan et al., 2021b), including the growing focus on data-centric AI (DCAI) (Jarrahi et al., 2022). Building from DCAI's focus on understanding the data used and produced throughout ML development, our framework sets a foundation for systematically analyzing social risks in data. If DCAI is focused broadly on shifting focus from the model to the data, our framework emphasizes human-centered angle within that focus. Some work in DCAI does bring attention to data sources and the sociocultural views they represent, such as those expressed through data annotation (Díaz et al., 2022; Arhin et al., 2021; Mishra & Gorana, 2021). However, this work has limited application to unstructured data.

An important part of dataset evaluation is determining when a data distribution is problematic. Future work in DCAI and RAI should explore the effects of different distributions on model performance and output bias. This work points to opportunities to use the framework to scaffold experiments to study the effects of different data distributions on model performance across contexts. In this way the framework is a concrete aid to what Jarrahi et al. (2022) calls "data benchmarking". Across iterative development of the same model, as well as across developments of distinct models, the framework can act as a consistent measuring stick for relating representations in data to model fairness.

At the same time, dataset evaluations are just one component of RAI evaluation. Much algorithmic fairness work focuses on data sources and dataset pre-processing; however, as Hooker (2021) argues, algorithm design choices, such as optimization for privacy guarantees, compression techniques, and even learning rate can contribute to model biases. Hooker also critically points out that dataset evaluations rely on a priori decisions about which features to evaluate and are inherently informed by human biases regarding what should or should not be prioritized. As a result, dataset evaluations must be considered alongside other approaches to mitigating risk and harm. For example, Hooker turns to model compression techniques to isolate data points at risk of exacerbated error rates as a way to guide further auditing. This enables iterative error analysis that DCAI calls for.

### 6.2 SUPPORTING RAI GOALS

RAI development requires data analyses that complement existing RAI processes while adapting to sociotechnical risks that are contextually determined. In response, the framework eschews automated test beds or fixed implementations (e.g., specific term lists or classifiers) and, instead, aims to standardize workflow planning. This includes structured guidance to repeat and localize analyses of human depictions in data. Our framework supports the development of transparency artifacts by standardizing results and by flagging benchmark tests to prioritize based on problematic data distributions. In this context, the framework stands as a structured auditing aid.

The same analysis results can warrant different actions depending on analysis goals. A primary motivation for this framework is to analyze data used for foundation models. High model training costs limit opportunities to run comprehensive studies to identify which mitigation strategies best support fairness and model performance. For RAI, this means making mitigation decisions with limited information about specific impacts. This challenge is exacerbated by data cascades, which can compound to produce out-sized, negative outcomes (Sambasivan et al., 2021b). Yet, the range of potential downstream risks warrants proactive decision making. While intervening on training is difficult when downstream applications are unclear, the framework can also be used for multimodal evaluations of fine-tuning data or model-generated data.

## 7 CONCLUSION

The open-ended nature of AI risks and harms poses challenges to RAI practitioners seeking to not only identify risks, but also take appropriate action to mitigate them, at times with limited information about how downstream models will be fine-tuned or applied. In response, we propose a standardized framework to evaluating unstructured datasets for downstream risk, with a focus on human representation. Building from critical dataset audits and other frameworks developed in recent years, we organize our framework around social representation and provide exemplar uses to demonstrate its application. Our framework is designed as a general-purpose starting point that is extensible to other modalities and application contexts, as needed.

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

# Appendix

## Table of Contents

## A  FRAMEWORK INTRODUCTION

Here we present a framework overview, detailed for text, image, and text-image application. Rather than an exhaustive list of every possible analysis, the Human-Content Associations section shows only social identity analyses for each modality. At the end of the overview we provide a list of example tools that can be used to meet the dependency requirements of the analyses. The full framework features recommended risk mitigation actions, however, in practice, mitigation steps should depend on development context and goals. The below tables list common analysis goals and mitigation actions and the following section on Guiding Considerations lists questions and prompts to guide framework use.

| Analysis Goals | |
| --- | --- |
| *Dataset Development* | Developing a dataset for training or evaluation through new data collection and/or adaptation of existing datasets |
| *Use Decisions* | Making decisions regarding appropriate use of a dataset, whether for training or evaluative purposes |
| *Model Understanding* | Investigating potential roots of or explanations for model behavior |
| *Auditing* | Auditing a dataset to fill documentation gaps, ensure legal or institutional compliance, or to foster greater public awareness |

Table 3: A non-exhaustive list of data analysis goals.

### A.1  GUIDING CONSIDERATIONS

- **What are the planned deliverables of the data being analyzed?**
  The planned outputs help to inform decisions regarding resource investment in analyses and potential mitigations. For example, a novel benchmark dataset may warrant more extensive analysis than a pre-training dataset because it will be repeatedly use to judge the performance of many models. Similarly, a fine-tuning dataset may be the subject of more modification than a pre-training dataset due to its smaller size and direct role in shaping model compliance with a range of fairness and policy constraints.

- **What are the primary goals the analyses will support?**
  Compliance audits and use decisions may only warrant documentation, or packaging and release actions; model understanding may simply be used to help diagnose model behavior

| Dev. Phase | Actions | Description |
|---|---|---|
| Data Collection/ Processing | Addition | Rebalancing distributions across an entire dataset or within specified categories with additional (potentially synthetic) data |
| | Removal | Filtering data to remove unwanted content |
| | Augmentation | Augmenting data, such as through data tagging (Anil et al., 2023) to allow a model to learn undesirable content while controlling its production downstream |
| | Flagging | Flagging analysis results for further downstream evaluation or documentation |
| | Non-Use | Not using the dataset, for example if applying analyses to different candidate datasets to decide which to use |
| Model Evaluation | Addt'l Benchmarking | Selection of additional evaluation benchmarks |
| | Benchmark Creation | Development of benchmarks to evaluate new concerns |
| Documentation | Warning | Documentation of general or use case-specific limitations |
| | Non-Use | Documentation of cases where the data should not be used |
| Packaging and Release | Licensing | Development of licensing and terms of use specifications |
| | Access | Development of limited access policies |

Table 4: A non-exhaustive list of actions that may be taken to address social risks identified in data.

(i.e., no mitigation action on data); whereas the development of a new dataset can involve many mitigation actions.

- **To what extent can completed or planned development steps be revisited or modified**
  Additional analyses and mitigations require time and resources, as does experimentation to compare which actions are most effective for a given project. For example, budget may be limited for additional data collection, while data filtering or tagging may still be possible.

- **To what extent is the dataset mutable? (i.e., can data be added, removed, or changed?)**
  The mutability of a dataset narrows the available mitigation actions and the degree to which data can be filtered or rebalanced.

- **Can new or additional model benchmarks be run?**
  When developing a new model, an ultimate consideration is whether biases identified in data will persist through training and finetuning steps. This is known to be true for a number of biases discovered in certain datasets. For novel or recently discovered biases, relevant benchmarks may not exist or are not feasible to create to test downstream models for the bias. This may place emphasis on other mitigation actions, depending on the risk associated with the identified bias.

- **Will the data or a model trained on the data be released or made available for external use? If so, can the release terms (e.g., terms of use, licensing, policies) be modified?**
  The use of data and models released externally cannot be as carefully controlled as data and models used strictly within a single institution. Safeguards against risks may be handled through terms of use or licensing agreements in order to mitigate data or model limitations.

- **Are there institutional or legal requirements regarding what should be analyzed in the data?**
  Legal and institutional compliance are a first step for planning analyses and reporting results. These can be supplemented with parallel fairness analyses of additional social groups, or analyzed more in depth to make mitigation decisions (e.g., analyzing which data sources feature the strongest identified biases in order to make filtering decisions)

- **Are there additional fairness or bias concerns a priori?**
  Beyond compliance concerns, consider analyses that involve groups or data subjects that have been previously impacted by similar systems or within which the context the data or model will be released.

- **What signals are available for analyzing social characteristics in the data?**
  The feasibility of running a given analysis depends on the available data features. Aside from straightforward considerations, such as whether a dataset contains faces that can be detected, varied social characteristics may map to a single data feature (e.g., common surnames across disparate racial and ethnic groups) or may not easily map to any data features (e.g., religion and disability are two examples of social characteristics that are not always 'visible' in images).

# B ANALYSIS OVERVIEW

## B.1 HUMAN FACTORS

### B.1.1 PRESENCE OF PEOPLE

- Personally-Identifiable Information
- Faces

### B.1.2 SOCIAL CHARACTERISTICS

- Social Identity Terms
- Pronouns
- Hateful Terms in Text
- Dialect
- Perceived Social Identity in Images
- Hateful Symbols in Images

## B.2 CONTENT FACTORS

### B.2.1 CONTENT

- Offensive Speech
- Topics
- Sexual Imagery
- Violent Imagery

### B.2.2 PROVENANCE

- Top Sources
- Source Geography
- Source Publication Time
- Language
- Data Duplication
- Dataset Overlap

## B.3 HUMAN X CONTENT ASSOCIATIONS

**Text**

- Social Identity Terms X Top Tokens
- Social Identity Terms X Offensive Speech
- Social Identity Terms X Topic

**Image**

- Perceived Social Identity Features X Top Image Features
- Perceived Social Identity Features X Sexual Imagery
- Perceived Social Identity Features X Violent Imagery

**Text-Image**

- Social Identity Terms X Sexual Imagery
- Social Identity Terms X Violent Imagery

- Perceived Social Identity Features X Text-Image
- Perceived Social Identity Features X Sexual Imagery
- Perceived Social Identity Features X Violent Imagery
- Perceived Social Identity Features X Offensive Speech
- Perceived Social Identity Features X Topic

## B.4 ANALYSIS DEPENDENCIES

Below is a list of technical dependencies required for framework analyses along with example classifiers that can be used.

**PII Detection:** Google Data Loss Prevention API (InfoType Detector)

**Face Detection:** Google Cloud Vision API (Image Annotation)

**Sexual Imagery:** Google Cloud Vision API (Safe Search Annotation)

**Violent Imagery:** Google Cloud Vision API (Safe Search Annotation)

**Object Classifier:** Google Cloud Vision API (Object Localization)

**Topic Classifier:** Google Cloud Natural Language API (Content Categories)

**Offensive Speech:** Perspective API (Note: Not offensive speech, but Toxicity)

**Language Classifier:** Google Cloud Natural Language API (Text Annotation

**Dialect Classifier:** No comprehensive classification approaches.

**Hateful Symbols:** No currently reliable method

**Social Identity Features:** A number of published audits point to important limitations of social identity classification systems (e.g., Buolamwini & Gebru (2018) and Scheuerman et al. (2019) audits of gender classification for image, and Raji et al. (2020a) audit of gender and age classification for image), therefore we encourage research and results interpretation with individual usage contexts in mind.

# C  LIST OF FRAMEWORK ANALYSES

## C.1  HUMAN FACTORS

Analyses focused on identifying individuals, social identity groups, and cultural groups in data.

**Presence of People**

### C.1.1  PERSONALLY IDENTIFIABLE INFORMATION (PII)

Personally identifiable information allows for an individual's identity to be potentially inferred through direct or indirect means. It includes, but is not limited to, names, email addresses, addresses, and images or video with identifying characteristics. The presence of personally identifiable information in a dataset is not inherently problematic. However, even if a dataset is not intended for public use, publishing a model trained on a dataset containing personally identifiable information risks "leaking" that information through attack methods designed to recover specific training instances from a model Carlini et al. (2021). Generative models that have overfit to the training distribution may also risk generating outputs that constitute a breach in privacy.

**Task:** Identify data containing PII

**Analysis Object:** Text | Inferred Text Signals

**Effort:** Medium

**Dependencies:** PII Detection

**Output:** Proportion and list of data points containing PII

**Action:** Flag PII to assess potential privacy violations that arise for model tasks or as a result of dataset publication

### C.1.2  PEOPLE IN IMAGES

As a subcategory of personally-identifiable information, faces in datasets are at risk of "leaking" through attack methods designed to recover specific training instances from a model Carlini et al. (2021). The presence of PII is not inherently problematic, however privacy risks must be evaluated in the context of specific tasks.

**Task:** Calculate the proportion of images that depict people (e.g., via face detection)

**Analysis Object:** Inferred Image Signals

**Effort:** Low

**Dependencies:** Face Detection

**Output:** Proportion of images that depict people

**Action:** Flag images of people

**Social Characteristics**

### C.1.3 Social Identity Terms

Under-representation in datasets can contribute to representational harms Barocas et al. (2017), such as strong under-representation of darker-skinned individuals in datasets linked to under-performance in pedestrian detection Wilson et al. (2019) and data imbalances generally causing issues for class prediction Johnson & Khoshgoftaar (2019). Attention to social identity representation in datasets also helps to support intersectional analyses which inherently focus on relatively smaller intersections of data but require sufficient data points to conduct.

**Task:** Calculate proportion of text referencing different social identity groups, considering unitary and intersectional groups

**Analysis Object:** Text

**Effort:** Low

**Dependencies:** Social Identity Term List

**Output:** Frequency of text depicting unitary and intersectional groups

**Action:** Flag social identity representation

### C.1.4 Pronoun Distribution

Pronoun distributions help provide a high-level snapshot of gender representation in datasets to evaluate their connection to model performance on gender bias benchmarks. Gender stereotypic associations have been well documented in NLP Bolukbasi et al. (2016); Caliskan et al. (2017) and mitigating imbalances in gendered terms in datasets has been shown to mitigate gender bias in benchmark performance Zhao et al. (2018).

**Task:** Count number of gender pronouns that occur

**Analysis Object:** Text

**Effort:** Low

**Dependencies:** Pronoun List

**Output:** Distribution of pronouns

**Action:** Flag pronoun distribution

### C.1.5 Hateful Terms in Text

When training datasets contain derogatory language, slurs, or offensive language, there is a risk models trained on the data will apply these terms in an inappropriate manner, e.g. describing people with such terms. Several recent audits of image classification datasets and image-to-text datasets have uncovered the presence racial and ethnic slurs and other derogatory and hateful language Birhane et al. (2021); Birhane & Prabhu (2021); Crawford & Paglen (2021).

**Task:** Count number of hateful terms that occur

**Analysis Object:** Text

**Effort:** Low

**Dependencies:** Hateful Term List

**Output:** Distribution of hateful terms, the groups they reference, and their occurrences

**Action:** Flag hateful terms

### C.1.6   DIALECT

Dialects vary across regions and cultures. Data that lacks manners of speaking or dialects prominently used by subgroups can lead to erasure of those dialects in resultant models, or categorization of non-dominant dialects as incorrect, low quality, or even offensive Sap et al. (2019).

**Task:** Calculate proportion of documents representing languages and dialects aimed to be supported in downstream applications

**Analysis Object:** Inferred Text Signals

**Effort:** High

**Dependencies:** Dialect Classifier

**Output:** Proportion of utterances representing different languages and dialects

**Action:** Flag dialect representation. Consider rebalancing strongly underrepresented dialects or qualifying downstream model capabilities.

### C.1.7   SOCIAL IDENTITY IN IMAGES

Training datasets that under-represent certain social identity groups can produce models that exhibit poor performance when presented with novel depictions of those groups. For example, image classifiers trained on datasets sourced predominantly from western countries have lower accuracy when applied to images from non-western countries Shankar et al. (2017), and facial analysis systems trained on datasets that skew heavily towards lighter skinned subjects have higher error rates when applied to images depicting faces with darker skin tones Buolamwini & Gebru (2018).

**Task:** Calculate proportion of images depicting social identity groups, considering unitary and intersectional groups

**Analysis Object:** Inferred Image Signals

**Effort:** Low

**Dependencies:** Perceived Social Identity Classifiers

**Output:** Proportion of images depicting unitary and intersectional groups

**Action:** Flag social identity representation

### C.1.8   HATEFUL SYMBOLS IN IMAGES

Visual imagery associated with hate groups (e.g., swastikas) may go undetected or undocumented in datasets. Identifying hateful content across modalities can help mitigate the risk of unintentionally generating hateful or offensive content. It is important to note that offensiveness of imagery is often dependent on cultural context, and hence any action to address this issue should account for this.

**Task:** Calculate proportion of images that depict known hateful symbols or text

**Analysis Object:** Inferred Image Signals

**Effort:** Not yet possible by automated methods

**Dependencies:** Hateful Symbol Classifier

**Output:** Proportion of images depicting known hateful symbols or text

**Action:** Flag images with hateful symbols

## C.2   CONTENT FACTORS

Analyses focused on identifying content that may heighten the sensitivity of human depictions.

**Content**

---

### C.2.1   OFFENSIVE SPEECH DISTRIBUTION

Identifying the distribution of offensive content in training data can help to understand the relationship between offensive speech in training data and model outputs as well as help mitigate the risk of unintentionally generating toxic content. Analyses of OWTC and OpenAI-WT suggest that toxic language generation has links to toxic language in web text data and toxic training examples that may be harder for a model to "forget" Gehman et al. (2020).

**Task:**  Calculate the distribution of toxicity

**Analysis Object:**  Inferred Text Signals

**Effort:**  Low

**Dependencies:**  Offensive Speech Classifier or similar (e.g., Perspective API[a])

**Output:**  Histogram of offensiveness across documents

**Action:**  Flag toxic content

---
[a]`www.perspectiveapi.com`

---

### C.2.2   TOPIC DISTRIBUTION

Topic distribution can provide a birds-eye view of the composition of the dataset across various topics such as sports, games, finance, etc. More importantly, it can also give an indication of sexually explicit or sensitive topics they contain, and potential biases they bring. For example, models trained primarily on news data have been shown to exhibit biases against particular country names and professions Huang et al. (2019). Assessments of data distributions across languages may highlight potential weaknesses across languages in generative tasks.

**Task:**  Calculate the distribution of topics

**Analysis Object:**  Inferred Text Signals

**Effort:**  Low

**Dependencies:**  Topic Classifier (e.g., Google Cloud Content Categories[a])

**Output:**  Histogram of topic distribution

**Action:**  Flag dominant topics

---
[a]`https://cloud.google.com/natural-language/docs/categories`

---

### C.2.3   SEXUAL IMAGERY

Many datasets have been identified as unintentionally containing sexually explicit content. For example, audits of image datasets such as Imagenet, TinyImage, and LAION-400M found pornographic and explicit imagery, predominantly depicting women Birhane et al. (2021); Birhane & Prabhu (2021). While some datasets may be explicitly curated to contain sexually explicit content, unintentional inclusion of such content can risk accidental generation of sexually explicit content by models trained on the data. Sexually explicit image content also raises potential ethical concerns regarding data sourcing since much publically available pornographic content has been stolen from sex workers or captured in a non-consensual manner Cole (2020).

**Task:**  Calculate the proportion of images that depict sexual content

**Analysis Object:**  Inferred Image Signals

**Effort:**  Low

**Dependencies:**  Visual Content Classifier (e.g., Cloud Vision API[a])

**Output:**  Proportion of images depicting sexual content

**Action:**  Flag sexual imagery. Consider filtering sexual imagery depending on downstream tasks and to avoid problematically-sourced dataset content

---

[a]https://cloud.google.com/vision/docs/reference/rest/v1/images/annotate#safesearchannotation

### C.2.4   VIOLENT IMAGERY

Like sexual imagery, violent content may feature in datasets as a part of efforts to moderate content, such as in the detection of cartoon content that may be inappropriate for children Khan et al. (2018). For general purposes, inadvertent generation of violent content may be undesirable or counter to institutional or platform policies.

**Task:**  Calculate the proportion of images that depict violent content

**Analysis Object:**  Inferred Image Signals

**Effort:**  Low

**Dependencies:**  Visual Content Classifier (e.g., Cloud Vision API[a])

**Output:**  Proportion of images depicting violent content

**Action:**  Flag violent imagery. Consider filtering violent imagery depending on downstream tasks

---

[a]https://cloud.google.com/vision/docs/reference/rest/v1/images/annotate#safesearchannotation

**Provenance**

### C.2.5   TOP SOURCES

The top source URL domains provide a high-level indication of the range of content most represented in web-scraped data. This, in turn, provides insight into document content, such as language and cultural content represented, as well as the prevalence of machine-generated text Dodge et al. (2021).

**Task:**  Count the top websites from which tokens were collected

**Analysis Object:**  Metadata

**Effort:**  Low

**Dependencies:**  None

**Output:**  List of the top websites ordered by tokens collected

**Action:**  Flag top URLs

### C.2.6 SOURCE GEOGRAPHIC SPREAD

Source URL domains provide an indication of geographic, cultural, and social representation. When training datasets have strong cultural skews, a resulting model can exhibit poor performance when presented with culturally or regionally specific depictions. For example, image classifiers trained on datasets sourced predominantly from western countries have lower rates of accuracy when applied to images from non-western countries Shankar et al. (2017).

**Task:** Calculate the known proportions of total tokens from different geographic regions (e.g., as determined through country-specific top-level domains such as, .uk, .de, .gh, etc.)

**Analysis Object:** Metadata

**Effort:** Low

**Dependencies:** None

**Output:** List of the top country-level domains with the proportion of data sourced from each

**Action:** Flag geographic spread. Consider rebalancing if regions relatively underrepresented in the dataset are prioritized for downstream uses.

### C.2.7 SOURCE DATA PUBLICATION TIME

The relevance of some data can change over time. For example, assessments of truth or offensiveness of a statement can change in relation to when it is evaluated. Data recency may have impacts on models supporting low-resource languages, which can disproportionately rely on religious or historical texts due to data scarcity Ahmadi & Masoud (2020).

**Task:** Identify the dates (years) of publication for data obtained

**Analysis Object:** Metadata

**Effort:** Low

**Dependencies:** None

**Output:** Histogram of data published per year

**Action:** Flag publication time. Consider rebalancing if prioritizing tasks likely to be relatively sensitive to data publication time (e.g., Q&A factuality; use of contemporary slang in generative language tasks)

### C.2.8 LANGUAGE

Languages vary across regions and cultures, however work in NLP frequently focuses on English, even when is data available for other languages Hovy & Prabhumoye (2021). As with under-representation of data and classes in other contexts, poorly represented languages in a language dataset may lead to worsened performance of a downstream model when applied to that langauge.

**Task:** Calculate proportion of documents representing languages aimed to be supported in downstream applications

**Analysis Object:** Text | Inferred Text Signals

**Effort:** Low

**Dependencies:** Language Classifier (e.g., Google Cloud Natural Language API)

**Output:** Proportion of utterances representing different languages

**Action:** Flag language representation. Consider rebalancing strongly underrepresented languages or qualifying downstream model capabilities.

### C.2.9  DATA DUPLICATION

Duplicate text has been identified in a number of datasets (e.g., Bandy & Vincent (2021)) and has been shown to worsen model performance on some tasks Allamanis (2019) as well as induce model memorization Lee et al. (2021), which can carry privacy risk Carlini et al. (2021).

**Task:** Calculate the proportion of duplicated data points

**Analysis Object:** Text | Image

**Effort:** Low

**Dependencies:** None

**Output:** Proportions of duplicated data points

**Action:** Filter duplicate data to reduce memorization and privacy risks and improve model performance

### C.2.10  DATASET OVERLAP

There is growing awareness and concern of test set contamination, i.e. presence of training instances in the test set, especially with large internet sourced datasets. Explicitly examining and mitigating any overlap in training and evaluation splits is increasingly commonplace (e.g., Trinh & Le (2018); Brown et al. (2020).

**Task:** Identify overlaps between the present dataset and relevant datasets and benchmarks

**Analysis Object:** Text | Image

**Effort:** Medium [Dependencies:] None

**Output:** For benchmark datasets identified, list of percent overlap with the current dataset based on exact matches of target text

**Action:** Filter dataset overlaps to preserve the validity of benchmark results

## C.3  HUMAN X CONTENT ASSOCIATIONS

Disaggregated analyses of how content is associated with human factors. Listed below are a selection of all possible associations between Human Factors and Content Factors. Disaggregated analyses enable evaluation of stereotypical or otherwise harmful associations between depictions of people and dataset content. For example, prior research has identified stereotype-aligned associations between gender and activities in image datasets Zhao et al. (2018); Hendricks et al. (2018), associations between muslims and violence in language models Abid et al. (2021), and associations between women of color and explicit content in text datasets Luccioni & Viviano (2021) and image-text datasets Birhane et al. (2021).

**Text**

### C.3.1  SOCIAL IDENTITY TERMS X TOP TOKENS

**Analysis Object:** Text

**Task:** For social identity terms in text, calculate top word co-occurrences for each

**Effort:** Low

**Dependencies:** None

**Output:** List of most frequent social identity terms and their respective top co-occurrences, demarcating identity terms (e.g., woman, Muslim, trans)

**Action:** Flag associations. Consider rebalancing harmful associations that emerge in model evaluations

### C.3.2 SOCIAL IDENTITY TERMS X TOPIC

**Task:** Calculate the distribution of topics within text, disaggregated by social identity terms or inferred social identity signals

**Analysis Object:** Inferred Text Signals + Social Identity Term List

**Effort:** Low

**Dependencies:** Topic Classifier (e.g., Google Cloud Content Categories[a])

**Output:** Distribution of topics disaggregated by social identity

**Action:** Flag disproportionate associations between social identity terms and stereotypical or sensitive topics

---

[a]`https://cloud.google.com/natural-language/docs/categories`

### C.3.3 SOCIAL IDENTITY TERMS X OFFENSIVE SPEECH

**Task:** Calculate the distribution of toxicity within text, disaggregated by social identity terms or inferred social identity signals

**Analysis Object:** Inferred Text Signals + Social Identity Term List

**Effort:** Low

**Dependencies:** Offensive Speech Classifier or similar (e.g., Perspective API[a])

**Output:** Distribution of toxicity, disaggregated by social identity

**Action:** Flag high toxicity. Consider rebalancing harmful associations that emerge in model evaluations

---

[a]`www.perspectiveapi.com`

**Image**

### C.3.4 PERCEIVED SOCIAL IDENTITY FEATURES X TOP IMAGE FEATURES

**Task:** Calculate co-occurrences between perceived social identity features and image features, such as objects and other people.

**Analysis Object:** Inferred Image Signals

**Effort:** Low

**Dependencies:** Object Classifier (e.g., Google Cloud Vision API[a])

**Output:** List of perceived social identity features and the top co-occurrences for each.

**Action:** Flag stereotypical associations. Consider rebalancing harmful associations that emerge in model evaluations.

---

[a]`https://cloud.google.com/vision/docs/object-localizer`

### C.3.5    PERCEIVED SOCIAL IDENTITY FEATURES X SEXUAL IMAGERY

**Task:** Calculate co-occurrences between perceived social identity features and sexual imagery.

**Analysis Object:** Inferred Image Signals

**Effort:** Low

**Dependencies:** Visual Content Classifier (e.g., Google Cloud Vision API[a]

**Output:** Proportion of images depicting sexual content disaggregated by social identity features

**Action:** Flag sexual content. Consider filtering sexual content depending on downstream tasks and to avoid problematically-sourced dataset content.

---

[a]`https://cloud.google.com/vision/docs/reference/rest/v1/images/annotate#safesearchannotation`

### C.3.6    PERCEIVED SOCIAL IDENTITY FEATURES X VIOLENT IMAGERY

**Task:** Calculate co-occurrences between perceived social identity features and violent imagery.

**Analysis Object:** Inferred Image Signals

**Effort:** Low

**Dependencies:** Visual Content Classifier (e.g., Cloud Vision API[a]

**Output:** Proportion of images depicting violent content disaggregated by social identity

**Action:** Flag violent content. Consider filtering sexual content depending on downstream tasks and to avoid problematically-sourced dataset content

---

[a]`https://cloud.google.com/vision/docs/reference/rest/v1/images/annotate#safesearchannotation`

### C.3.7    PERCEIVED SOCIAL IDENTITY FEATURES X HATEFUL SYMBOLS

**Task:** Calculate co-occurrences between perceived social identity features and hateful symbols detected in images.

**Analysis Object:** Inferred Image Signals

**Effort:** Not currently possible

**Dependencies:** Hateful Symbol Classifier

**Output:** Proportion of images depicting sexual content disaggregated by social identity

**Action:** Flag hateful content. Consider filtering hateful content depending on downstream tasks.

**Text-Image**

### C.3.8    SOCIAL IDENTITY TERMS X SEXUAL IMAGERY

**Analysis Object:** Social Identity Term List + Inferred Image Signals

**Task:** Calculate co-occurrences between social identity terms and sexual imagery.

**Effort:** Low

**Dependencies:** Visual Content Classifier (e.g., Cloud Vision API[a]

**Output:** Proportion of images depicting sexual content disaggregated by social identity term

**Action:** Flag associations. Consider rebalancing harmful associations that emerge in downstream model evaluations

---

[a]`https://cloud.google.com/vision/docs/reference/rest/v1/images/annotate#safesearchannotation`

### C.3.9  SOCIAL IDENTITY TERMS X VIOLENT IMAGERY

**Analysis Object:**  Social Identity Term List + Inferred Image Signals

**Task:**  Calculate co-occurrences between social identity terms and violent imagery.

**Dependencies:**  Visual Content Classifier (e.g., Cloud Vision API[a])

**Output:**  Proportion of images depicting violent content disaggregated by social identity term

**Action:**  Flag associations. Consider rebalancing harmful associations that emerge in downstream model evaluations

**Effort:**  Low

---

[a]https://cloud.google.com/vision/docs/reference/rest/v1/images/annotate#safesearchannotation

### C.3.10  PERCEIVED SOCIAL IDENTITY FEATURES X TOP TEXT TOKENS

**Analysis Object:**  Inferred Image Signals + Text

**Task:**  Calculate co-occurrences between perceived social identity signals and tokens in associated text.

**Effort:**  Low

**Dependencies:**  Perceived Social Identity Classifiers

**Output:**  List of most frequent co-occurrences for each social identity signal

**Action:**  Flag associations. Consider rebalancing harmful associations that emerge in downstream model evaluations

### C.3.11  PERCEIVED SOCIAL IDENTITY FEATURES X OFFENSIVE SPEECH

**Analysis Object:**  Inferred Image Signals + Inferred Text Signals

**Task:**  Calculate co-occurrences between perceived social identity signals and tokens in associated text.

**Effort:**  Low

**Dependencies:**  Offensive Speech Classifier or similar (e.g., Perspective API[a])

**Output:**  Distribution of toxicity, disaggregated by social identity

**Action:**  Flag high toxicity. Consider rebalancing harmful associations that emerge in model evaluations

---

[a]www.perspectiveapi.com

### C.3.12  PERCEIVED SOCIAL IDENTITY FEATURES X TOPIC

**Analysis Object:**  Inferred Image Signals + Inferred Text Signals

**Task:**  Calculate co-occurrences between perceived social identity signals and tokens in associated text.

**Effort:**  Low

**Dependencies:**  Topic Classifier (e.g., Google Cloud Content Categories[a])

**Output:**  Distribution of toxicity, disaggregated by social identity

**Action:**  Flag high toxicity. Consider rebalancing harmful associations that emerge in model evaluations

---

[a]https://cloud.google.com/natural-language/docs/categories

