# OpenReview forum: "A Conceptual Framework for Analyzing Social Representation in Unstructured Data"
_ICLR.cc/2024/Conference — Submitted to ICLR 2024_

### Official Review · Reviewer_9nGw · 2023-10-31

**Soundness:** 2 fair
**Presentation:** 3 good
**Contribution:** 2 fair
**Rating:** 5
**Confidence:** 4

**Summary:**

This paper presents a framework to facilitate analysis of unstructured data regarding how it represents humans and how this might lead to downstream harms. They demonstrate how this framework may be used in the context of two well-established datasets, C4 and LAION-400M. The framework is focused specifically on evaluating social representations of people by categorizing analyses into (1) who is in the data, (2) what is in the data, and (3) associations between the two. The authors also provide background on previous work related to dataset transparency, audits, and Responsible AI. For (1), the authors consider the factors of "presence of people" (including personally-identifying information) and "social characteristics" (which the authors acknowledge must be adapted to local context).
For (2), the authors include the aspects of "content" and "provenance." The authors also include components of "Output" (the results of an analysis), "Action" (mitigation actions to be taken), "Analysis Object", "Effort", and "Dependencies" (which correspond meta-properties of the analysis method itself.)

**Strengths:**

The authors provide a good survey of different aspects of harm related to socially-salient information in the training data of models. The idea of looking at associations between human and content factors is also a neat way to conceptualize these issues and connect data-related concerns to problems like stereotype amplification and the propagation of exclusionary norms.

Regarding mitigation, the authors provide a strong list of key questions for an author to consider.

Overall, it is a good synthesis/survey of existing work on mitigating social bias from a data-centric perspective.

**Weaknesses:**

1. Inconsistent framing of context/use case: The abstract frames the paper in the context of "foundation model development" but fails to make this distinction throughout the paper.
2. Too vague, to the point of limited utility: throughout the paper, the authors assert that specific measures cannot be delineated. For example, they write that social characteristics "measurement must be adapted to local context and time" and because they vary, "static identity term list cannot be exhaustive." The authors also do not make a distinction between different modalities of data, e.g. audio vs. visual vs. text, which have very different ways of having traces of social data that are worthy of being discussed. Yet this vagueness seems directly at odds with the authors' goal of developing a framework that is easily applicable. The paper suggests that it serves as a "starting point" (Section 5), but this seems insufficient given that most of these ideas are from other existing works.
2.1. In Section 3.1.1, the suggestion of "faces or bodies in visual data or voice in audio data" are good suggestions but do not come close to being thorough in the range of possibilities, which makes it limited in utility. For example, https://arxiv.org/pdf/2309.15084.pdf (Figure 2 on page 4) is a good example of a more thorough survey of different types of human data that occur specifically in visual data. A similar type of typology would strengthen this work.
3. In "Social Characteristics," the authors use the primary examples of "perceived" social characteristics that "must be inferred, frequently using predictive methods." This sidelines the many ethical issues around predicting social characteristics and reproduces these harms: see
a) Lockhart, Jeffrey W., Molly M. King, and Christin Munsch. "Name-based demographic inference and the unequal distribution of misrecognition." Nature Human Behaviour (2023): 1-12. b) Cao, Yang Trista, and Hal Daumé III. "Toward gender-inclusive coreference resolution." arXiv preprint arXiv:1910.13913 (2019). c) So, Richard Jean, and Edwin Roland. "Race and distant reading." PMLA 135, no. 1 (2020): 59-73. d) Hanna, Alex, Emily Denton, Andrew Smart, and Jamila Smith-Loud. "Towards a critical race methodology in algorithmic fairness." In Proceedings of the 2020 conference on fairness, accountability, and transparency, pp. 501-512. 2020. Given that social representations of people are a central focus of this work, such ethical concerns should be much more centered in this paper.
4. Given the broad utility that the authors aspire toward, the case studies are not particularly convincing of how the framework is easily or widely applicable.
5. The example of evaluating age representation in C4 and queer representation in LAION-400M is interesting, but seems to presuppose a lot of existing knowledge about what concerns/harms the authors are looking for.

**Questions:**

Who is the imagined audience for this paper?
In Section 3.1.3, you refer to "human and content factors." Does "human" encompass all of the "who" is in the data and "content" encompass all of the "what" is in the data? If so, why is content just a subset of the latter in Section 3.1.2?
What do you mean by "Responsible AI (RAI)"? This term is continuously used without being defined.

---

### Official Review · Reviewer_qstp · 2023-11-06

**Soundness:** 2 fair
**Presentation:** 2 fair
**Contribution:** 2 fair
**Rating:** 3
**Confidence:** 3

**Summary:**

This paper introduces a framework for planning responsible AI (RAI) workflows, focusing on data assessment, documentation, and risk mitigation. Existing guidance for evaluating ML workflows is scant, often leading to repetitive or incomplete analyses. Additionally, the high cost of model training constrains extensive research into which mitigation strategies optimally balance fairness and performance.

**Strengths:**

- Raises awareness of the importance of analyzing social representation in machine learning's unstructured data.
- Introduces a universal framework to identify the representation of individuals or groups in unstructured data.
- Suggests actionable measures for addressing representational issues during various development phases.
- Demonstrates the framework's effectiveness through systematic social risk analyses of the C4 and LAION-400M datasets.

**Weaknesses:**

- Lacks empirical evidence on the framework's risk mitigation effectiveness for downstream tasks.
- Absence of comparative analysis with other methods tackling similar issues.
- Despite acknowledging the variability of "good" social representation, the framework does not adapt to different contexts or provide standard risk measurement metrics.
- There is no demonstration of the framework's impact on risk perception before and after its application.
- While the framework poses high-level, human-centered questions, it falls short of offering standardized metrics for practitioners to use for development and iterative assessment.

**Questions:**

none

---

### Official Review · Reviewer_HJFp · 2023-11-07

**Soundness:** 1 poor
**Presentation:** 1 poor
**Contribution:** 1 poor
**Rating:** 3
**Confidence:** 3

**Summary:**

The paper is about a framework for analyzing social representation in unstructured data to make use and documentation decisions. The proposed framework is to guide analysis in understanding human representation (from a responsible AI perspective) in unstructured data and identify downstream risks. The authors propose hypothetical action steps in service of dataset use, development and documentation.

**Strengths:**

- The paper is about an interesting topic, building a framework for data analysis from a responsible AI perspective.
- The authors cover various related works in the paper.
- The authors show some use cases on toy large unstructured datasets, C4 and LAION-400M.

**Weaknesses:**

- The paper is difficult to read. The writing style and the structure can be improved. The big picture and the aim of the framework are discussed a lot, but there are not a lot of details discussed. Figure 1 needs to be more detailed on the actual components of the framework.
- The novelty in the methodology is limited. It is not clear what are the newly proposed ideas and methodology in the proposed framework.
- This paper might be a better fit for a more applied track and a conference/workshop/journal with a focus on such applications/frameworks.
- The experimental setup is not very clear. It would be easy to follow if the authors can add the dataset details in a table, provide with the research questions and a quantitative way to evaluate the results.
- The proposed framework is more discussed in the paper as a hypothesis and how it will work theoretically, than in practice showing how it works according with results.

**Questions:**

- It will be useful if the authors can add their contribution in the introduction section.
- It will be useful if the authors can provide the challenges and limitations to existing methods of this problem.
- It will be useful if the authors can add a comparison to other similar frameworks, and find a way to evaluate the performance of the proposed framework.
- It will be useful if more datasets and use cases are presented in the experimental setup.
- There is no comment on whether the framework/code is going to be publicly available.

**Details Of Ethics Concerns:**

-

---

### Official Review · Reviewer_Nevx · 2023-11-08

**Soundness:** 2 fair
**Presentation:** 3 good
**Contribution:** 2 fair
**Rating:** 3
**Confidence:** 4

**Summary:**

This paper presents a framework for systematically assessing unstructured datasets to identify potential biases and risks that can affect the fairness of machine learning models. The framework categorizes analyses into three sections: "Who is in the Data," "What is in the Data," and "Human × Content Associations", and identifies issues related to social identity representation, content characteristics, and their associations. The paper provides two toy examples to demonstrate the application of the framework. In the first example, the framework is used to evaluate age representation in a dataset, while the second example focuses on evaluating queer representation and sexual content in another dataset.

**Strengths:**

- The paper introduces a novel framework for evaluating unstructured datasets from a responsible AI perspective.
- The framework emphasizes the importance of documentation and transparency in dataset evaluation, aligning with the broader trends in responsible AI research. The paper provides a couple of practical examples of how the framework can be applied to real-world datasets.
- The paper is overall well-written with a comprehensive appendix.

**Weaknesses:**

- The paper primarily focuses on qualitative assessments of data and provides limited quantitative metrics. Quantitative measures are important in enhancing the rigor of dataset evaluation and making the framework more comprehensive. Focusing on qualitative analysis makes the conceptual framework, as presented, time-consuming to apply comprehensively, especially to large datasets. This suggests further refinement is needed to make it more scalable for practical use.
-  The paper lacks empirical validation of the framework on a diverse set of datasets and applications, which are necessary to demonstrate the modality-agnostic nature of the framework. Demonstrating its effectiveness through real-world case studies, showcasing how easy it is to extend to extensible to new modalities and combinations, and providing comparative analysis with similar frameworks would strengthen the contributions.
- While the paper identifies issues in datasets, it offers limited discussion on effective mitigation strategies. For example, there is a risk of introducing unintended bias when applying the framework, as the choices made in the analysis and actions can themselves be influenced by the researchers' perspectives. It is unclear how practitioner decisions affect the insights stemming from the application of this framework.
- Technical contributions are limited. The conceptualization of a general framework for social identity representation under three dimensions (Who, What, Human × Content) has merit and the potential to serve as foundations for researchers and practitioners working on responsible AI, but the focus on generality across datasets, modalities, and use cases, make the proposed work vulnerable to oversimplification. Different datasets and applications may require nuanced and context-specific approaches, which the framework might not fully capture. It would be beneficial to provide more specific guidelines or techniques for conducting the analyses and actions proposed in the framework.


The work could benefit from further refinement, the inclusion of quantitative analysis to facilitate scalability, and more detailed guidance on mitigation strategies, in particular addressing potential bias in the framework's application.

**Questions:**

Several questions come to mind, please try to address the ones that are most valuable:

- Could you provide more details on the specific components of the framework, such as "Analysis Object," "Effort," and "Dependencies"? How are these components determined, and can you provide examples of their application in real-world scenarios? How do you recommend practitioners decide on the most appropriate mitigation actions?

- Have you conducted any empirical validation of the framework on more diverse real-world datasets or applications? What challenges do you anticipate in implementing the framework in practical scenarios?

- How does the framework address scalability issues when applied to such datasets, and what considerations should practitioners keep in mind when scaling the analyses and actions?

- The paper promotes a modality-agnostic approach, but datasets and applications can vary significantly. How do you recommend adapting the framework to specific contexts and ensuring that it captures the nuances of different data types and use cases?

- How does the proposed framework integrate with or complement existing fairness and responsible AI practices, such as bias mitigation techniques and fairness-aware algorithms? Does it align with established standards and guidelines? Have you sought feedback from the AI research and practitioner community during the development of this framework?

---

### Official Review · Reviewer_x2iJ · 2023-11-10

**Soundness:** 2 fair
**Presentation:** 3 good
**Contribution:** 2 fair
**Rating:** 3
**Confidence:** 3

**Summary:**

The paper "A Conceptual Framework for Analyzing Social Representation in Unstructured Data" addresses the challenges of analyzing unstructured data in the development of foundational machine learning models. The authors propose a framework to standardize the evaluation of such data, with a focus on the representation of human identities and the risks associated with their downstream applications, acknowledging the significant impact of data on AI behavior. The framework is designed to guide practitioners in responsibly planning their AI workflows, which includes data evaluation, documentation, and risk mitigation, and is applicable across various data modalities. To demonstrate the framework's utility, the authors apply it to two case studies using large web text corpora, identifying potential risks and suggesting actions for responsible dataset use. The paper contributes to Responsible AI by addressing the gap in systematic methods for conducting fairness audits in unstructured data.

**Strengths:**

1. Originality: The paper makes an original contribution by proposing a new framework for analyzing social representation in unstructured data. The originality of the paper is evident in the integration of ideas from dataset auditing and fairness evaluation to tackle the complexities of unstructured data.

2. Clarity: The writing and structure of the manuscript are clear, providing a coherent flow from the introduction of the problem to the presentation of the framework. The authors effectively explain complex concepts in a manner that could be accessible to a broad audience, which is commendable.

3. Significance: This framework could become a significant tool for researchers and practitioners working towards more ethical AI systems if further developed and validated.

**Weaknesses:**

1. Methodological Development: The methodology behind the framework lacks depth, particularly regarding its applicability to the wide variety of unstructured data that exists. This concern is especially relevant given the increasing complexity and scale of datasets to be utilized in deep learning models.

2. Validation and Empirical Evidence: The primary weakness of the paper lies in its validation process. The utilization of toy examples, while helpful for illustrative purposes, does not suffice to establish the efficacy of the proposed framework across the varied landscape of unstructured data that practitioners face. For robust validation, the framework should be subjected to a battery of tests across datasets of different scales, complexity, and real-world messiness. This would not only reinforce the reliability of the framework but also demonstrate its practical applicability and adaptability. Moreover, the current lack of quantitative analysis means there is no clear measure of performance improvement or bias mitigation when using the framework, which is crucial for empirical validation. This rigorous level of empirical evidence is necessary to validate any claims of improved fairness and efficacy.

3. Comparative Analysis: A significant drawback of the paper is the absence of a detailed comparative analysis with current state-of-the-art frameworks and methodologies. Such a comparison is vital to critically evaluate the advantages, limitations, and added value of the new framework. The paper should articulate how and why the proposed framework outperforms existing methods or, alternatively, how it could be integrated with current approaches to provide a more comprehensive tool. Specific benchmarks, metrics, and scenarios where the proposed framework excels (or falls short) compared to others would greatly enhance its credibility and relevance.

4. Actionability and Implementation Guidance: The action steps are conceptually insightful but lack the necessary detail for practical implementation. For practitioners to adopt the framework, they require concrete guidance on its integration into existing workflows. This should include specific recommendations on handling different types of unstructured data, along with examples of the framework in action. Case studies that demonstrate successful integration and outcomes are needed to bridge the gap between theory and practice.

5. Impact Measurement: The manuscript does not adequately address how to quantitatively measure the impact of the framework on AI fairness and bias mitigation. Presenting specific metrics and assessment methodologies is essential for validating the framework's effectiveness in practical applications.

**Questions:**

1. Can you provide specific metrics that were used or could be used to quantitatively measure the effectiveness of your framework in mitigating bias?
2. Is there scope for developing a comparative performance analysis against existing methods using these metrics?
3. Could you include a more detailed comparative analysis with current state-of-the-art frameworks?
4. How would you propose to measure the impact of your framework on the fairness of AI systems in practice?
5. How does the framework perform in terms of computational efficiency when applied to large datasets?

---

### Official Review · Reviewer_MXER · 2024-01-02

**Soundness:** 2 fair
**Presentation:** 3 good
**Contribution:** 3 good
**Rating:** 5
**Confidence:** 1

**Summary:**

.

**Strengths:**

.

**Weaknesses:**

.

**Questions:**

.

**Details Of Ethics Concerns:**

.

---

### Author Response · Authors · 2023-11-14
**Thank you**

Thank you very much to each of the reviewers for your feedback and suggestions, and we appreciate your interest in this topic of work! Given the extent of the revisions required for this work, we will focus on using this feedback to improve the manuscript for a future conference or journal submission.

---

### Meta-Review · Area_Chair_wWSA · 2023-12-05

**Metareview:**

The paper aims to develop a method for analyzing human identity representation  in AI. The ideas have merit, but substantial revisions are required due to insufficient validation, lack of comparative analysis, vague implementation guidelines, and overlooked ethical considerations.

**Justification For Why Not Higher Score:**

Substantial revisions are required due to insufficient validation, lack of comparative analysis, vague implementation guidelines, and overlooked ethical considerations.

**Justification For Why Not Lower Score:**

N/A

---

### Decision · Program_Chairs · 2024-01-16

Reject